# Patient Engagement Using Telemedicine in Primary Care during COVID-19 Pandemic: A Trial Study

**DOI:** 10.3390/ijerph192214682

**Published:** 2022-11-09

**Authors:** María Asunción Vicente, César Fernández, Mercedes Guilabert, Irene Carrillo, Jimmy Martín-Delgado, José Joaquín Mira

**Affiliations:** 1Health Psychology Department, Miguel Hernandez University, 03202 Elche, Spain; 2Foundation for the Promotion of Health and Biomedical Research of Valencia Region (FISABIO), 0313 Alicante, Spain; 3Instituto de Investigación e Innovación en Salud Integral, Facultad de Ciencias Médicas, Universidad Católica de Santiago de Guayaquil, Guayaquil 090603, Ecuador; 4Alicante-Sant Joan d’Alacant Health Department, 03013 Alicante, Spain

**Keywords:** chronic conditions, COVID-19, telemedicine, m-Health, engagement, adherence

## Abstract

The correct treatment of most non-transmissible diseases requires, in addition to adequate medication, adherence to physical activity and diet guidelines, as well as health data monitoring and patient motivation. The restrictions caused by the COVID-19 pandemic made telemedicine tools and mobile apps the best choice for monitoring patient compliance. The objective of this study was to analyze the benefits of an m-Health solution designed specifically for chronic patients during the COVID-19 pandemic. A pragmatic clinical trial with pre–post measurements of a single group was carried out with 70 patients (aged 40+) with one or more chronic conditions. Patients were provided with an ad hoc mobile app and health data measuring devices according to their diseases. The health status of the patients was monitored remotely by health professionals who could also modify the patient’s objectives according to their evolution. The results obtained show an average fulfillment of objectives of 77%. Higher fulfillment values: medication adherence (98%) and oxygen saturation (82%); lower fulfillment values: weight (48%), glucose (57%), and distance walked (57%). Globally, the ad hoc app was rated 8.72 points out of 10 (standard deviation 1.10). Concerning the pre–post analysis, there were significant improvements vs. prior apps used by the participants in the following items: improved physical activation and better control of blood pressure, diet, weight, glucose, and oxygen saturation. In conclusion, the telemedicine tool developed was useful in increasing patient engagement and adherence to treatment.

## 1. Introduction

### 1.1. Chronic Diseases and Telemedicine

Chronic diseases cause an important limitation to the quality of life, productivity, and functional status of patients. They represent a heavy burden in terms of morbidity and mortality and a challenge to the organizational capacity and sustainability of health systems.

Chronic diseases are associated with high levels of therapeutic non-compliance. Some factors include the complexity of the therapeutic regimen and intrinsic patient factors, such as negative beliefs towards some drugs or greater fragility. Moreover, medication errors at home are surprisingly frequent. A study by Meyer-Massetti et al. [1], where drug-related problems (DRPs) in home care were analyzed through a database search, concluded that the frequency of potentially inappropriate medications (PIMs) ranged from 19.8% to 48.4% with 26% of PIMs being severe. Moreover, patients and caregivers were responsible for 42.3% of DRPs. Previously, Doran [2], through a cohort study in different Canadian regions, estimated that over 2% of hospital admissions were caused by medication-related errors in home care.

The patient’s commitment to self-care has been shown to contribute both to therapeutic compliance and the correct use of medications. Self-care is an essential factor that integrates the knowledge, ability, and desire of patients to commit to achieving better results [3]. It involves learning to control the evolution of the disease; knowing how to manage its physical, emotional, and social impact; and deciding how and where to obtain the health care that is needed at all times. This commitment has been shown to be directly related to improvements in health status (including lower mortality), motivates professionals, and contributes to a rational consumption of health resources [4]. Previous studies have tried to promote patients’ commitment to the therapeutic objectives agreed upon with their professionals (including health habits and clinical variables). The PROPRESE program [5] proposed the personalization and further monitoring of patient objectives for chronic cardiometabolic patients in primary care. The results obtained showed statistically significant improvements versus the control group in different cardiovascular risk factors.

Commitment to self-care, together with the availability of health resources on the internet, led to the creation of the term e-patient, or people who seek online guidance for their own illnesses or those of their friends and family [6]. Nowadays, e-patients benefit from a variety of telemedicine technologies, such as live video communication with professionals, remote patient monitoring, and mobile health applications.

Different studies have analyzed the impact of e-patients’ attitudes on healthcare quality, including a study by Kim and Kwon [7] that concluded that health information literacy programs are required for patients to be able to make informed decisions on their own health. Focusing on chronic illnesses and personal health records, the study presented in [8] draws a similar conclusion: the need to provide education to consumers and providers to increase the utility of personal health records. From the professionals’ perspective, Masters et al. [9] carried out a questionary study where surgeons positively valued electronic communication with patients, as well as patient activities such as bringing material from the internet to the consultation.

Concerning patient commitment and telemedicine, the VALCRONIC program [10] proposed the continuous monitoring of cardiometabolic patients by means of specialized hardware. Adherence to medication has also been addressed in previous studies, such as the project ALICE [11], where a medication self-management mobile app showed to obtain significant improvements both in adherence and a reduction in medication errors. In general, telemedicine applications are particularly well-suited to a wide variety of chronic conditions. The number of m-Health solutions (health apps, web applications, wearable devices) aimed at patients with more prevalent chronic conditions is growing due to the population that has smart devices (mainly smartphones and tablets) increasing, as well as technological advances.

However, while most patients are satisfied with telemedicine, its acceptability and effectiveness are a source of skepticism among professionals [12]. Most studies suggest that telemedicine is cost-effective, activates patients, and allows them to remain in their natural environment for as long as possible compared to traditional care [13,14]. The best results are achieved with patients from rural areas that have more difficulty accessing traditional care [15,16,17]. However, there are some results that invite new studies to verify which elements of these programs are responsible for the improvements identified [18]. For example, it is necessary to address whether telemedicine achieves greater patient commitment to therapeutic goals, healthy lifestyles, adherence, and safe use of medication at home. Most studies are focused on verifying the effectiveness of m-Health applications in terms of improvements in clinical parameters, such as the recent review by Lewinski et al. [19], where they identified 8662 studies related to the effect of telemedicine use in chronic conditions. However, only five studies were valid for data extraction (none of them dealt with COPD or chronic obstructive pulmonary disease). A common result in those five studies was a very low certainty that telehealth influenced hospital admissions, emergency attendance, or clinical parameters.

Apart from comparing the clinical results of telemedicine vs. in-person care, there are other issues that require further research, such as how to achieve patient engagement in a digital environment. A study carried out by Barello et al. [20] concludes that a change in focus, properly analyzing the patient's needs, is required to increase the potential of telemedicine for patient engagement.

### 1.2. COVID-19 Pandemic

Chronic conditions suffered particularly during the COVID-19 pandemic, where the attention was focused on COVID-19 patients, overwhelming primary care and making it almost impossible for patients of all other illnesses (including chronic conditions) to access health services. Although this problem was more severe during the first waves, the access barrier created persisted through all pandemic waves [21]. Telemedicine has been highlighted as a valid alternative to provide care in optimal conditions by means of different applications (videoconferencing, remote monitoring, or mobile health, among others). The goals are to increase efficiency, to reach patients unable to access health facilities, and, in general, to reduce in-person visits as much as possible [22,23]. Post-pandemic, telemedicine continues to produce optimal care, although different challenges will need to be addressed [24].

In this context, chronic conditions require novel strategies to reduce the frequency of in-person hospital or primary care visits, including tools for: (1) improving therapeutic adherence, (2) avoiding medication errors, (3) increasing engagement in self-care, and (4) remote monitoring.

### 1.3. Proposal and Context

We analyzed the benefits of a complete m-Health solution designed specifically for a diverse group of chronic patients suffering from diabetes, hypertension, hypercholesterolemia, heart disease, or COPD. This m-Health solution responds to the needs of the COVID-19 pandemic, as the tools developed were particularly useful for reducing in-person hospital or primary care visits. Different from other studies, the intervention was carried out in actual clinical practice and not in controlled experimental conditions.

The intervention was designed for primary care (family practice), which is a key element in the patient's journey through the Spanish health system. Basically, when patients need medical assistance, they must first visit their primary care physician, who will give them advice or treatment and, if necessary, will send them to other professionals. Primary care physicians are the gatekeepers responsible for authorizing access to specialty care, hospital care, and diagnostic tests (obviously, emergency situations bypass the primary care visit). Each patient is assigned a primary care professional so that further visits will always be scheduled with the same professional for better follow-up of each patient. The intervention was designed for primary care as an extra tool for the patient–professional relationship.

The use of telemedicine applications in the Spanish health system is still limited but keeps growing. The study presented in [25] reflects that in Spain, as of 2018, online patient visits accounted for 2.61% of all visits and that remote monitoring reached 10.29%. More recently, the COVID-19 pandemic forced a sudden increase in the adoption of e-Health systems, and even after the pandemic peaked, adoption rates kept increasing, and this trend seems to be continuing into the future [26]. The intervention proposed fits this scenario, where new e-Health technologies and tools need to be evaluated in real environments before their implementation.

## 2. Methods

A pragmatic clinical trial (carried out under identical conditions to those of clinical practice) was carried out, with pre–post measurements of a single group.

The study protocol was approved by the Research Ethics Committee of the Sant Joan University Hospital in Alicante (reference 18/324, 27 June 2018). The study was registered with ClinicalTrials, reference NCT04159558.

### 2.1. Setting, Subjects, and Recruitment

The study was carried out in primary care in seven health centers of the Valencian Community. This care level is ideal for implementing actions that increase patient activation and promote healthy habits, patient compliance, and safety.

Subjects: patients over 40 years old who agreed to participate in the study after informed consent and with one or more chronic conditions, such as diabetes, hypertension, hypercholesterolemia, heart disease, or COPD. Patients with neurological or mental illness were excluded.

Based on the previous studies ALICE [11] and VALCRONIC [10], a single group of 70 participants was determined. Considering a worst case of 5% of losses, the minimum effect size that could be detected in pre–post analyses (Wilcoxon signed-rank test, alpha = 0.05, power = 0.95) was 0.42, so even small effect sizes could be detected.

For recruitment, a simple random sampling was applied for all the patients who attended the health centers between December 2020 and June 2021. The physician responsible for patient care was given instructions to register the patients in the experiment after informed consent. Recruitment of 4–5 patients per week was estimated, which implied a recruitment duration of 30 weeks.

### 2.2. Intervention and Ad hoc App 

On one side, the intervention was based on the PROPRESE program (PROgram for Cardiovascular SEcondary PREvention) [5] designed to encourage the patient’s commitment to the therapeutic goals agreed upon with their doctor. This program has demonstrated its ability to involve the patient in self-care and its contribution to secondary prevention in cardiovascular processes. It is an intervention program that establishes personalized objectives for each patient based on risk stratification and level of health knowledge, dedicating more effort to patients with the highest risk and least knowledge. The intervention presented in this paper, in line with the PROPRESE program, addressed the agreement of health goals between patients and health professionals. These health goals could evolve during the intervention, always through agreement. Health goals that can be agreed upon between patient and doctor and that can be modified as the patient evolves include, among others:Activity: minimum number of steps or distance walked per day.Weight range (maximum and minimum values).Blood pressure (SBP and DBP).Diet.Oxygen saturation.Blood glucose level.

On the other side, the intervention was also based on the previous ALICE project [11] designed to improve adherence and reduce medication errors. The functionalities of the previous ALICE project (medication reminders) were included, as well as additional tools to allow patients to hear prerecorded information about their medicines by scanning the medicine boxes.

All the previously mentioned functionalities (among others) were included in an ad hoc health app (named *TeleHealth*), which was supplied to patients, as well as peripherals, according to their needs. Doctors were able to remotely access all information needed to monitor the achievement of their patient’s goals.

The system architecture of the *TeleHealth* app was composed of three main elements: patient devices, patient mobile app, and professional control panel. The patient app was developed for the Android operating system, and all patient health data were managed through a Google Fit account of the patient.

#### 2.2.1. Patient Devices

These devices were used to measure patient activity and health data. Among these devices, there were activity wristbands, weight scales, blood pressure monitors, glucometers, pulse oximeters, and heart rate monitors. Some of these devices were connected to the Google Fit account of the patients, so data uploading was automatic. For unconnected devices, the patients were asked to input data manually in their apps (daily or with lower frequencies). Patients lacking the necessary devices had the devices provided to them.

#### 2.2.2. Patient Mobile App

The patient app has multiple functionalities. It (1) allows patients to input health data (data not uploaded automatically through their devices) and shows the evolution of these data (today, yesterday, weekly average, monthly average) along with the patient objectives; (2) rewards patients fulfilling their objectives and show personalized encouragement messages, depending on their evolution and fulfillment of objectives; (3) helps patients with their medicines, showing previously recorded basic information (when to take it, how to take it, what is it for, etc.); and (4) a homogeneous alarm system to remind the patients to take their medicines, start exercising, or take their health measurements.

#### 2.2.3. Professional Control Panel

Each patient had a health professional in charge of monitoring her/his health status, keeping track of health alerts, and adjusting her/his objectives. Professionals were obviously required to log in with their personalized access data and were able to visualize the evolution of all their patients in terms of (1) activity (steps and distance walked) and diet; (2) health data (weight, blood pressure, glucose, heart rate, oxygen saturation); and (3) medication adherence. All these data were monitored along with the personalized objectives for each patient. Moreover, the professionals received warning messages whenever some of the data registered for a patient were out of the safety margins. Depending on the evolution of each patient, the professional could modify her/his personalized objectives remotely.

The complete architecture of the system is shown in Figure 1.

Some screenshots of the patients’ mobile app and professionals’ control panel are shown in Figure 2 and Figure 3, showing, as an example, the management of blood pressure control.

Figure 2 shows the patients’ view of the app. From left to right: (a) an alarm fires to tell the user that it is time to take the daily blood pressure measure. The patient can accept this message or ask to be notified later. (b) After accepting the message, the patient introduces her/his blood pressure values. Some helpful tips on how to take the correct measurements are available if needed. If the blood pressure values are found to be potentially dangerous, the patient will be provided with a set of extra questions to check possible symptoms and evaluate the risk. Depending on the answers, a suggestion of visiting a doctor or calling emergency services will be shown. (c) Once the value is inputted, and assuming that the values are within safe ranges, the patient visualizes a summary of the evolution of her/his blood pressure and the fulfillment of the objectives agreed upon with the doctor: today’s values, yesterday’s values, as well as weekly and monthly averages. (d) If the patient needs more details, a complete plot of the blood pressure evolution and goal values is also available, with variable time ranges.

**Figure 1 ijerph-19-14682-f001:**
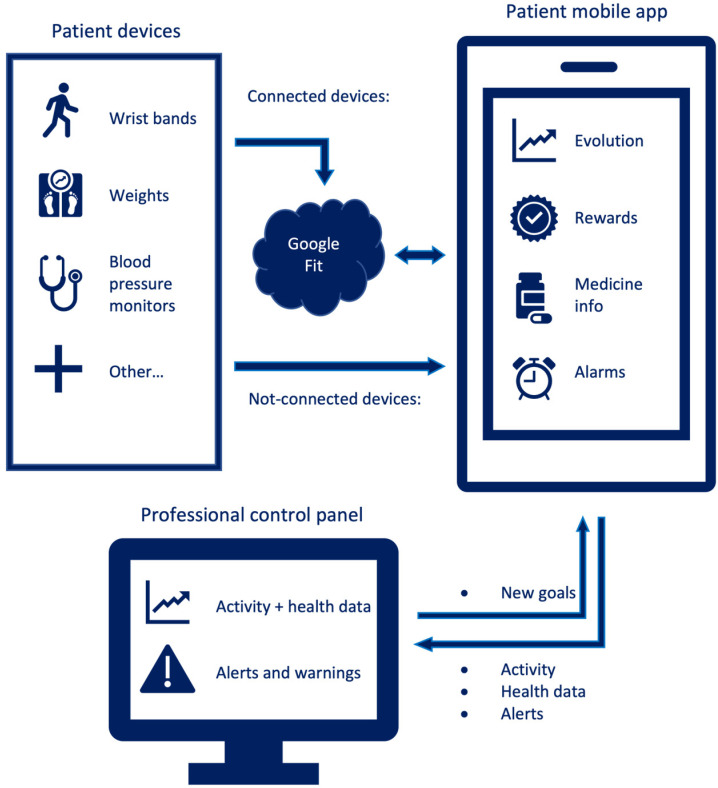
Architecture of the *TeleHealth* system.

**Figure 2 ijerph-19-14682-f002:**
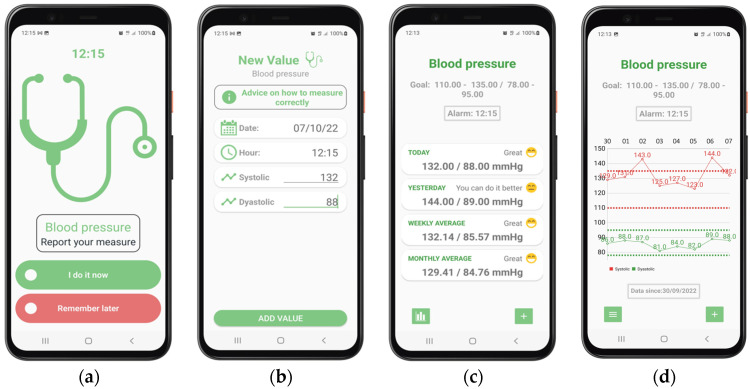
Some screenshots from the *TeleHealth* system—patient’s view (mobile app): (**a**) Blood pressure alarm; (**b**) Blood pressure data input; (**c**) Blood pressure evolution summary; (**d**) Blood pressure evolution detail. Higher-resolution screenshots are available as Appendix A.

**Figure 3 ijerph-19-14682-f003:**
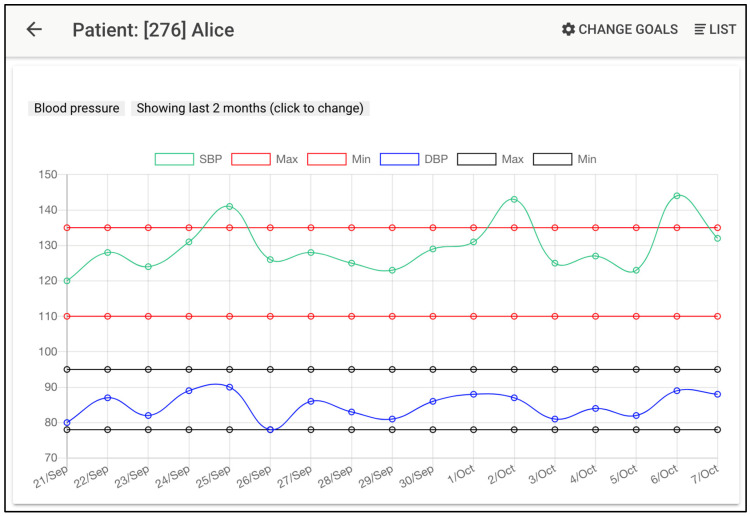
*TeleHealth* system: professional’s view (control panel). Available as Appendix A for better visualization, together with additional screenshots.

Figure 3 shows the professional’s view (control panel), where they can remotely monitor their patients’ data. This screenshot shows the evolution of the patient’s blood pressure, showing values and allowed ranges according to the agreed goals. The data that the doctors can visualize include:(1)A patient form data, including current diseases.(2)A complete list of the patient’s health data values (height, blood pressure, diet, etc.), with clear indications of which objectives are being fulfilled and clear warning signs for dangerous values.(3)If the doctor decides to obtain more details about the evolution of certain data, a complete plot is available, showing values and allowed ranges according to the agreed-upon goals (e.g., a screenshot of Figure 3 for blood pressure).(4)A detailed day-by-day list of certain health data also showing a set of security questions that are answered by the patient when her/his values are potentially dangerous. Again, a clear indication of dangerous symptoms is shown.

Appendix A include a high-resolution screenshot for each of the previous items.

### 2.3. Measures and Statistical Analysis

The intervention was designed as a pre–post experiment, with no control group. The data gathered during the experiment include:Demographic data for all participants (age, sex, self-perceived health status, diagnosis, health data goals, as well as experience with mobile apps and ICTs).A pre–post questionnaire to check the improvements obtained after the intervention period. A Wilcoxon signed-rank test was used to detect significant changes.A daily report of health data vs. goal values (steps and distance walked, weight, diet, blood pressure, glucose, heart rate, oxygen saturation), as well as a daily report of treatment adherence. A further stepwise linear regression between the fulfillment of objectives and demographic data was carried out to detect significant effects.A final satisfaction questionnaire to rate the *TeleHealth* app. Gender effects were checked using exact Fisher tests (for yes/no questions) and Chi-squared tests (for rating questions).A specific questionnaire to check the usefulness of the *TeleHealth* app for avoiding medication errors.

All statistical analyses were carried out using SPSS software version 28.0.0.0 (190).

## 3. Results

Data were obtained from 70 patients (36, 51.4% males). The average age was 60.9 years (max. 75, min 44, std. 7.1). A total of 8 of 70 (11.4%) lived alone, but 66 of 70 (94.3%) were responsible for managing their medication. A total of 17 of 70 patients (24.3%) took more than 5 types of drugs per day. Although in most cases, their knowledge of ICTs was limited, 63 of 70 (90%) had been using a mobile phone for over 10 years, and 25 of 70 (35.7%) had used a health app previously. The characteristics of those who participated are summarized in Table 1.

### 3.1. Fulfillment of Health Objectives

Patients agreed with their doctors on their health goals. Thus, the health items to control and the acceptable value ranges were specific for each patient.

Globally, there was an average fulfillment of objectives of 77%. The higher fulfillment values were found in medication adherence (98%) and oxygen saturation (82%), while lower fulfillment values were found in weight (48%), distance walked (57%), and glucose (57%) (see Table 2).

All health objectives were analyzed to find possible relations between the percentage of fulfillment and demographic data using stepwise linear regression. The “steps walked” data were dependent on age (older age was associated with lower fulfillment) and the number of weeks using the app (longer usage was associated with higher fulfillment). The detailed results are shown in Table 3.

### 3.2. Effects of TeleHealth

One of the goals of the intervention was to avoid medication errors. A questionnaire was used to check the patients’ perceived usefulness of the app in these aspects.

For patients that previously used pillboxes (*n* = 8), the first question was, “Was the app more useful than the pillbox you used previously?”, and all answers (100%) found the app more useful.

Of all patients (whether they used other medication tools previously or not, *n* = 70), 63 patients (90%) answered positively to the following set of questions:The app helped me to avoid medication mix-ups.The app helped me to organize doses according to my doctor’s indications.The app helped me not take incompatible medicines simultaneously.The app helped me to take the dose recommended by my doctor at the correct time.

A pre–post test was also used to measure the self-perceived improvement after using our ad hoc app *TeleHealth*. The results show significant improvements in all health data included in the questionnaire: physical activation, blood pressure control, diet, weight, glucose, and oxygen saturation. The detailed results of a Wilcoxon signed-rank test are shown in Table 4.

### 3.3. Satisfaction of Participants

A total of 68 participants rated the app and answered a short questionnaire related to their satisfaction with different app functionalities.

Globally, the app was rated with an average mark of 8.72 out of 10 (max: 10, min: 6, standard deviation: 1.104). A gender effect was found (Chi-squared test, *p* = 0.046), as female patients gave higher marks (average of 8.94) than male patients (average of 8.50). Table 5 details the results obtained.

Concerning each functionality separately, the most valued aspects were ease of use (97.1% of participants); health data line charts (88.2%); and the feeling of safety due to the continuous monitoring of doctors through the app (66.2%); while the less-valued aspects were the capability to report app errors (8.8%); the alarms (8.8%); and the help section, including video-tutorials (17.6%).

There were two extra questions related to the global experience of the app. In this sense, 97.1% of participants would recommend the *TeleHealth* app to their friends or family, and among those who used health apps previously, 56.5% would consider using *TeleHealth* instead of their previous apps.

Possible gender effects were analyzed through exact Fisher tests for all the questions, but no effects were found. The complete data are shown in Table 6.

## 4. Discussion

Telemedicine makes it possible to reduce the need for face-to-face consultation, but it does not completely replace it. One of the main objectives is to achieve adequate therapeutic compliance, for which there have already been previous developments since as early as the work of Haynes [27] or, more recently, the PROPRESE project [5] and the VALCRONIC project [10], which also focused on the remote monitoring of these patients.

The *TeleHealth* mobile application has proven to be useful in reducing the number of consultations and maintaining adequate control of patients. In this sense, 66.2% of patients felt safer knowing that their doctors could monitor their health through the app (see Table 6). However, a significant percentage of patients (33.8%) responded negatively to the same question, which may indicate that these people would feel safer with regular face-to-face visits.

In the current study, the only aspect where a gender effect was found was global satisfaction with the app (female patients were more satisfied). No other gender effects were found, neither for the fulfillment of health objectives nor for the level of satisfaction with specific aspects of the *TeleHealth* program. However, in the literature, the level of acceptability of telemedicine interventions has been found to be dependent on gender, as the review presented in [12] reflects, as well as in other demographics such as age or socio-economic status. Further research on these aspects can result in future telemedicine applications being more adapted to different population sectors.

The COVID-19 pandemic has accelerated the implementation of telemedicine applications to reduce in-person visits, particularly during its worst episodes [22,23]. The momentum gained by these applications due to the pandemic is supposed to be maintained in future years, even if the pandemic disappears completely. The *TeleHealth* app presented in this paper is in line with this tendency: reducing in-person hospital or primary care visits, even during normality periods.

The fast development of novel telemedicine applications requires a similarly fast evolution of legislation on the topic. Even though there are differences between countries, legislation on telemedicine is lacking or missing in multiple aspects [28]. Among these important aspects that should be regulated, we can cite confidentiality issues or liability for malpractices.

There is also a requirement for the elimination of certain barriers that limit the widespread implementation of telemedicine applications. The review presented in [29] analyses these barriers over 22 previous studies and detects, as the most common problems, the technical challenges for the staff (appearing in 11% of the studies), the resistance to change (8%), the reimbursement (5%), the age of patients (5%), and the level of education of patients (5%). These barriers should be addressed by adapting current and future telemedicine applications and by performing specific interventions on healthcare professionals and patients.

Concerning the different chronic diseases, not all of them benefit from telemedicine at the same level. As an example, the review presented in [30] analyses the benefits for diabetes patients (through the data obtained from 31 previous studies) and concludes that patients with type 2 diabetes benefit more than patients with type 1 diabetes when lowering HbA1c levels using telemedicine applications.

Patient and provider satisfaction with telemedicine is extremely relevant. The review presented in the work of Nanda and Sharma [22] (25 previous studies, 48,144 surveyed patients, 146 providers in 12 different countries) concludes that the main benefits of telemedicine reported by patients are (1) time saved due to less traveling, (2) no need to wait in queues, (3) cost efficiency, (4) convenience, and (5) accessibility, while health providers were also satisfied with teleconsultation and did not find telemedicine to be associated with higher workloads. In our study, patients also found the use of telemedicine beneficial, as shown in the post-experiment questionnaire answers (see Table 5 and Table 6).

Concerning the fulfillment of health objectives and considering that the experiment was carried out in a real environment, the results obtained must be considered relevant to determine whether similar interventions should be generalized. As mentioned in the Results section, there was an average fulfillment of 77%, and the only fulfillment below 50% was that of the weight control (48%). However, weight varies slowly, and this relatively low value may have improved in a longer-duration intervention. Globally, the intervention should be considered successful in terms of the improvement in health objectives.

Among the limitations of the study, there are some elements to consider:

First, the experiment was carried out in a real context of primary care (family practice). Although primary care is a key element in the patient journey through the Spanish health system, some of the results may not be comparable to those obtained in other contexts (e.g., other medical specialties, hospitals, and health systems of other countries).

Second, the limited number of participants and the absence of a control group does not allow us to extrapolate the results obtained. Further experiments with a larger and more representative number of participants, including a control group, are required to confirm these results.

Third, this study was focused on measuring self-perceived improvements and not the quality of life or organ function improvements. That would require a completely different intervention over a longer period.

Fourth, a comparative cost analysis should also be performed to assess the feasibility of telemedicine programs compared to in-person health services. Future studies need to address this feasibility study.

## 5. Conclusions

Patient engagement, which was a key element of chronic conditions during the COVID-19 pandemic, can be improved with specifically designed telemedicine applications, such as that developed for the current study. Improvements were found in the self-perceived control of physical activity, weight, diet, blood pressure, glucose level, and oxygen saturation when using this tool.

Patient–doctor agreement on health data goals allows patients to achieve high fulfillment values, particularly in adherence to medication, as well as oxygen saturation and blood pressure measures.

According to our study, there are no significant gender effects in the fulfillment of health objectives or the level of satisfaction with specific aspects of the *TeleHealth* program. Only global satisfaction with the app was higher among female patients. These results should be studied further as they are not in line with previous research.

## 6. Patents

The *TeleHealth* app was registered through Fisabio (“Fundación para el Fomento de la Investigación Sanitaria y Biomédica de la Comunitat Valenciana” or “Foundation for the Promotion of Health and Biomedical Research of the Valencian Community”, http://fisabio.san.gva.es/ (accessed on 15 June 2022) with reference code FS_FOR048, and register number E_102_2020 06.10.2020.

## Figures and Tables

**Table 1 ijerph-19-14682-t001:** Demographic data of participants.

Data	*n* (%)
Sex	
Male	36 (51.4)
Female	34 (48.6)
Self-perceived health status	
Bad	4 (5.7)
Average	26 (37.1)
Good	40 (57.1)
Cohabitants	
Lives with wife/husband	42 (60)
With some other relative	20 (28.6)
Live alone	8 (11.4)
Diagnosis ^1^	
Smoking habit	6 (8.6)
Hypertension	33 (47.1)
Hypercholesterolemia	12 (17.1)
Diabetes	16 (22.9)
Coronary heart disease	4 (5.7)
COPD	7 (10)
Take more than 5 different medicines per day	17 (24.3)
Responsible for managing their own medication	66 (90)
Used pillboxes previously	8 (11.4)
Used mobile phones for more than 10 years	63 (90)
Have used a health app previously	25 (35.7)
Patients with treatment goals agreed with doctor ^1^	
Oxygen saturation	8 (11.4)
Blood pressure	56 (80)
Glucose	19 (27.1)
Weight	40 (57.1)
Heart rate	23 (32.9)
Steps walked	56 (80)
Diet	28 (40)
App usage frequency ^2^	
Irregular	19 (27.1)
Regular	15 (21.4)
Continuous	36 (51.4)

^1^ Patients may have more than one diagnosis and more than one treatment goal agreed. ^2^ App usage frequency: irregular (less than twice a week), regular (twice a week), and continuous (more than twice a week).

**Table 2 ijerph-19-14682-t002:** Fulfillment of health objectives.

Health Objective	Fulfillment %
Adherence to treatment	98
Oxygen saturation	82
Diastolic blood pressure	81
Systolic blood pressure	75
Diet	64
Heart rate	64
Steps walked	60
Glucose	57
Distance walked	57
Weight	48
Average	77

**Table 3 ijerph-19-14682-t003:** Stepwise linear regression between fulfillment of steps walked and demographic data.

Data	Beta	95% CI		*p*
Sex	−2.0	−21.0	16.9	0.830
Age	−1.7	−3.1	−0.2	0.026
Cohabitants	−7.2	−22.5	8.0	0.346
Bad self-perceived health status	−16.6	−53.0	19.8	0.364
Average self-perceived health status	−8.8	−27.85	10.3	0.361
Years using mobile phones	−0.1	−1.5	1.4	0.957
App usage frequency	−2,8	−8.5	2.9	0.335
Weeks using the app ^1^	0.9	0.2	1.7	0.016

^1^ Number of weeks actively using the app (uploading steps walked).

**Table 4 ijerph-19-14682-t004:** Wilcoxon signed-rank test for the pre–post questionnaire (*n* = 70).

Question	Pre, *n* (%)	Post, *n* (%)	*p*
Does your smartphone help you in controlling your blood oxygen saturation?	0 (0)	8 (11.4)	0.005
Does your smartphone help you in improving your level of physical activity?	6 (8.6)	50 (71.4)	<0.001
Does your smartphone help you in controlling your blood pressure?	2 (2.9)	52 (74.3)	<0.001
Does your smartphone help you in improving your weight?	2 (2.9)	31 (44.3)	<0.001
Does your smartphone help you in controlling your glucose levels?	2 (2.9)	18 (25.7)	<0.001
Does your smartphone help you in sticking to a healthy diet?	0 (0)	16 (22.9)	<0.001

**Table 5 ijerph-19-14682-t005:** Satisfaction of participants.

Mark (0–10)	Total	Female	Male
6	2	1	1
7	7	4	3
8	20	7	13
9	18	6	12
10	21	16	5
Total	68	34	34

**Table 6 ijerph-19-14682-t006:** Satisfaction of participants after using the *TeleHealth* app.

Question	Total, *n* (%)	Female, *n* (%)	Male, *n* (%)	*p*
I found the app easy to use daily	66 (97.1)	32 (94.1)	34 (100)	0.493
Health data line charts were useful in following the goals agreed upon with my doctor	60 (88.2)	30 (88.2)	30 (88.2)	1.0
I feel safer knowing that my doctor monitors my health continuously through the app	45 (66.2)	23 (67.6)	22 (64.7)	1.0
I have used the video tutorials and help section	12 (17.6)	6 (17.6)	6 (17.6)	1.0
Alarms were useful in fulfilling my health data goals	6 (8.8)	3 (8.8)	3 (8.8)	1.0
Notifying errors from within the app was useful	6 (8.8)	2 (5.9)	4 (11.8)	0.673
I would recommend to my friends or family the use of this app to improve their health	66 (97.1)	33 (97.1)	33 (97.1)	1.0
From now on, I will use *TeleHealth* instead of other apps I used previously for controlling my health ^1^	13 (56.5)	4 (40.0)	9 (69.2)	0.222

^1^ Only those using apps previously, *n* = 23 (10 female). For all other rows, *n* = 68 (34 female).

## Data Availability

The anonymized study data are available for other researchers through a request to the corresponding author.

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
