# Peer review of "Patient Engagement Using Telemedicine in Primary Care during COVID-19 Pandemic: A Trial Study"

_ijerph, 2022, doi:10.3390/ijerph192214682_

Round 1

Reviewer 1 Report

First of all I would like to congratulate the authors for investigating in this field as necessary as unknown, besides generating an application that can be very useful.

Regarding the research itself, in general terms the manuscript is well constructed, however there are a number of essential changes for its publication and some topics that should be incorporated on and tried to address to improve the text as a whole. They are detailed below and alternative ways to tackle them are proposed.

Both the methodology and the results are well presented and I do not think there is much more to detail. However, an important problem is that this type of research, which is not representative due to its sample size and configuration, usually has a control group that is not exposed to the stimulus, to measure not only effectiveness in the participants themselves, but also to verify these effects against a group of people who follow normal treatment or follow-up (unchanged).

If it were possible and the authors had data on adherence in other (non-exposed) patients, it would be of immense value to include these data in the results sections as well as in the discussion and conclusion, giving a better account of the real effect of the application.

If this information is not available, it is equally important to point out this fact within the limitations of the project, considering the possible extension to other groups or to larger and more representative samples in the future.

Consequently, I suggest that the authors remove any reference to or qualification of "statistically significant" in the conclusions, thereby lowering the claims made, as there is no representative sample or control group at this stage. In any case, the results are remarkable and stand up for themselves, but I think it is not appropriate to use terms that are generally linked to representativeness and inference from the sample to the general population.

Specific changes:

In the second page lines 55 to 63 I think the autors are surrounding a concept that they should present and they problably know, as it’s e-Patient. Some possible references about them can be:

-        Ferguson, T., & Frydman, G. (2004). The first generation of e-patients. British Medical Journal, 328(7449), 1148–1149. https://doi.org/10.1136/bmj.328.7449.1148

-        Masters, K., Loda, T., Johannink, J., Al-Abri, R., & Herrmann-Werner, A. (2020). Surgeons’ interactions with and attitudes toward E-patients: Questionnaire study in Germany and Oman. Journal of Medical Internet Research, 22(3). https://doi.org/10.2196/14646

-        Kim, K., & Kwon, N. (2010). Profile of e-patients: Analysis of their cancer information-seeking from a national survey. Journal of Health Communication, 15(7), 712–733. https://doi.org/10.1080/10810730.2010.514031

-        Gee, P. M., Paterniti, D. A., Ward, D., & Miller, L. M. S. (2015). e-Patients perceptions of using personal health records for self-management support of chronic illness. CIN - Computers Informatics Nursing, 33(6), 229–237. https://doi.org/10.1097/CIN.0000000000000151

-        Ramos, A. C., Buceta, B. B., da Silva, Á. F., & Lorenzo, R. B. (2020). Ehealth in Spain: Evolution, current status and future prospects | La esalud en España: Evolución, estado actual y perspectivas de futuro. Saude e Sociedade, 29(4), 1–12. https://doi.org/10.1590/S0104-12902020190886

In the entire introduction there is no data or reference to place either the context of the case of analysis (it should briefly and concisely point out how health care is provided in Spain, how the proposed pilot fits into the system) or the general eHealth situation in this country. The section "1.2 Telemedicine" is probably the right place to introduce such a context. Some references to situate this information could be:

-        Ramos, A. C., Buceta, B. B., da Silva, Á. F., & Lorenzo, R. B. (2020). Ehealth in Spain: Evolution, current status and future prospects | La esalud en España: Evolución, estado actual y perspectivas de futuro. Saude e Sociedade, 29(4), 1–12. https://doi.org/10.1590/S0104-12902020190886

-        Sust, P. P., Solans, O., Fajardo, J. C., Peralta, M. M., Rodenas, P., Gabaldà, J., Eroles, L. G., Comella, A., Muñoz, C. V., Ribes, J. S., Monfa, R. R., & Piera-Jimenez, J. (2020). Turning the crisis into an opportunity: Digital health strategies deployed during the COVID-19 outbreak. JMIR Public Health and Surveillance, 6(2). https://doi.org/10.2196/19106

Author Response

Thank you for your constructive review, we have modified the paper according to your suggestions. Please find below the specific changes introduced in the paper, point by point:

1) Lack of control group

As you mention, there is no control group. The experiment was designed exclusively as a pre-post analysis.

According to your request, we have clearly stated the absence of control group in different points through the paper (methods section and discussion section). Consequently, all references to statistically significant results have been omitted in the methods, results, and conclusions sections.

Additionally, the discussion section now proposes as future work the extension of the intervention to other, larger, groups of participants, to have a more representative sample, including a control group.

2) E-patient term.

As pointed out by the reviewer, this term was not explicitly included in the original version of the paper.

The new version includes the term in the introduction section, together with a concise explanation, and includes the references proposed by the reviewer.

3) Spanish context where the intervention is proposed.

The original version of the paper did not describe in detail this context.

The new version includes information on the introduction section detailing the Spanish Health System context (both in general terms and focusing on telemedicine), and also the possible fit of the proposed intervention. The references proposed by the reviewer have been added.

Apart from the points above, a thorough grammar and style check has been carried out to improve the readability of the paper.

Reviewer 2 Report

The purpose of this study was to analyze the benefits of a complete m-Health solution specifically designed for patients with multiple chronic conditions such as diabetes, hypertension, hypercholesterolemia, heart disease, or chronic obstructive pulmonary disease. m-Health is a mobile app and health data measurement device specifically designed for patients. It helps health professionals remotely monitor the health status of their patients, and they can also change the patient's goals based on the patient's development. The purpose of this study was to analyze the benefits of a complete m-Health solution specifically designed for patients with multiple chronic conditions such as diabetes, hypertension, hypercholesterolemia, heart disease, or chronic obstructive pulmonary disease. m-Health is a mobile app and health data measurement device specifically designed for patients. It helps health professionals remotely monitor the health status of their patients, and they can also change the patient's goals based on the patient's development.

1)The citations in this paper are divided into two parts: chronic disease and telemedicine. However, the content of the two parts is relatively combing, and there is no strong progressive relationship between the literature, and the citations of the literature are less.

2)The intervention designed in this study was based on the application developed by the PROPRESE plan and the ALICE project, which was carried out in actual clinical practice rather than under controlled experimental conditions, and the results of the study had certain limitations.

3)The evaluation indicators of the research results in this paper include the evaluation of patients' health outcomes and satisfaction, and the evaluation indicators of economic latitude after the implementation of related projects are insufficient. The indicators of health outcomes included self-perceived physical activity, body weight, diet, blood pressure, blood glucose level and blood oxygen saturation, which mainly focused on basic physiological indicators. Organ function and quality of life for each disease were few and simple, making it difficult to comprehensively evaluate the effect of the program.

4)This paper mentions that m-Health can change patients' goals according to their development, but the paper lacks specific evidence to support it, so it is suggested to explain in detail.

Author Response

Thank you for your detailed review, we have modified the paper according to your suggestions. Please find below the specific changes introduced in the paper, point by point:

1)The citations in this paper are divided into two parts: chronic disease and telemedicine. However, the content of the two parts is relatively combing, and there is no strong progressive relationship between the literature, and the citations of the literature are less.

The introduction section has been restructured to improve the flow from chronic conditions to telemedicine, with added references and concepts, like the e-patient concept.

Besides, the context of the experiment (Spanish health system) has been detailed in the introduction section for a better understanding of the fit of the proposed intervention.

2)The intervention designed in this study was based on the application developed by the PROPRESE plan and the ALICE project, which was carried out in actual clinical practice rather than under controlled experimental conditions, and the results of the study had certain limitations.

The link with previous projects PROPRESE and ALICE is now rewritten to improve clarity. The new version of the paper states more clearly the study limitations in the discussion section (including the limited number of participants and the absence of a control group).

Additionally, the discussion section now proposes as future work the extension of the intervention to other, larger, groups of participants, to have a more representative sample, including a control group.

3) The evaluation indicators of the research results in this paper include the evaluation of patients' health outcomes and satisfaction, and the evaluation indicators of economic latitude after the implementation of related projects are insufficient. The indicators of health outcomes included self-perceived physical activity, body weight, diet, blood pressure, blood glucose level and blood oxygen saturation, which mainly focused on basic physiological indicators. Organ function and quality of life for each disease were few and simple, making it difficult to comprehensively evaluate the effect of the program.

As the reviewer points out, the study is focused on measuring self-perceived improvements and not quality of life or organ function improvements. The new version also states this point in the discussion section.

4) This paper mentions that m-Health can change patients' goals according to their development, but the paper lacks specific evidence to support it, so it is suggested to explain in detail.

The change of patient’s goals is a key point in the contract (or agreement) between patient and professional. According to the patient evolution, both parts may agree on a modification of these goals.

The new version of the paper details this fact to a greater extent in the methods section.

Apart from the points above, a thorough grammar and style check has been carried out to improve the readability of the paper.